# Antioxidant, Nutritional Properties, Microbiological, and Health Safety of Juice from Organic and Conventional ‘Solaris’ Wine (*Vitis vinifera* L.) Farming

**DOI:** 10.3390/antiox13101214

**Published:** 2024-10-09

**Authors:** Ireneusz Ochmian, Sebastian W. Przemieniecki, Magdalena Błaszak, Magdalena Twarużek, Sabina Lachowicz-Wiśniewska

**Affiliations:** 1Department of Horticulture, West Pomeranian University of Technology Szczecin, Słowackiego 17 Street, 71-434 Szczecin, Poland; 2Department of Entomology, Phytopathology and Molecular Diagnostics, University of Warmia and Mazury in Olsztyn, 10-719 Olsztyn, Poland; sebastian.przemieniecki@uwm.edu.pl; 3Department of Bioengineering, West Pomeranian University of Technology Szczecin, Słowackiego 17, 71-434 Szczecin, Poland; magdalena.blaszak@zut.edu.pl; 4Department of Physiology and Toxicology, Faculty of Biological Sciences, Kazimierz Wielki University, 85-064 Bydgoszcz, Poland; twarmag@ukw.edu.pl; 5Department of Medical and Health Sciences, Calisia University, 4 Nowy Świat Street, 62-800 Kalisz, Poland; s.lachowicz-wisniewska@akademiakaliska.edu.pl; 6Department of Biotechnology and Food Analysis, Wroclaw University of Economics and Business, 53-345 Wroclaw, Poland

**Keywords:** grapes, antioxidant activity, organic farming, microbiome, plant protection, mycotoxins, polyphenols

## Abstract

This study investigated the technological parameters, microbiological, and functional properties of juice from Solaris grapes grown under conventional and organic farming systems to assess how these cultivation methods influence juice quality. The one-year study focused on key aspects such as the levels of health-promoting polyphenols, the presence of mycotoxins, and pesticide residues. Organic grapes showed greater bacterial and fungal diversity, with significant differences in dominant genera. *Sphingomonas* and *Massilia* were the predominant bacteria across both systems, while *Erysiphe* was more common in conventional grapes, and *Aureobasidium* was abundant in both. Despite the presence of genes for mycotoxin production, no mycotoxins were detected in the juice or pomace. Organic juice exhibited significantly higher levels of polyphenols, leading to enhanced antioxidant properties and improved technological characteristics, including lower acidity and higher nitrogen content. However, residues of sulfur and copper, used in organic farming, were detected in the juice, while conventional juice contained synthetic pesticide residues like cyprodinil and fludioxonil. These findings highlight that while organic juice offers better quality and safety in terms of polyphenol content and antioxidant activity, it also carries risks related to residues from organic treatments, and conventional juice poses risks due to synthetic pesticide contamination.

## 1. Introduction

The fruit’s surface serves as a habitat for microorganisms, and grapes, like other fruits, possess a distinct chemical composition. The chemical composition of grapes is characterized by a high content of simple sugars and a low pH, which provides an environment conducive to the growth of yeasts and bacteria that facilitate sugar fermentation. Clusters of grapes are extensively populated by various yeast species, including *Hanseniaspora, Candida, Metschnikowia, Pichia, Rhodotorula, Torulaspora,* and *Aureobasidium*, as well as bacteria such as *Lactobacillus, Acinetobacter, Pseudomonas, Leuconostoc, Sphingomonas, Lactococcus, Bacillus, Arthrobacter*, and others [1,2]. Additionally, there are mold fungi and microorganisms from other habitats [3].

The microbiological composition of grapes is particularly important in the production of natural wines, where fermentation occurs spontaneously through a consortium of indigenous yeast and bacterial strains. Consequently, the unique secondary bouquet of wines is largely dependent on the microbiological composition of the grapes. Under conditions of elevated humidity and temperature, filamentous fungi, most often belonging to the genera *Alternaria, Botrytis, Cladosporium, Fusarium, Cladosporium, Rhizopus*, and *Mucor,* tend to grow [4]. Fungal activity depends on weather conditions; in dry growing seasons, where the grapes remain dry, fungi do not develop but persist in spore form. Certain species of mold fungi are toxigenic (for example, from the genera *Aspergillus, Penicillium*, and *Fusarium*) and pose a potential risk to consumers of fruits, juices, or wines. The most common contaminants of fruits are ochratoxins, patulin, aflatoxins, fumonisin, and alternariol [5]. The question arises as to whether limited chemical protection, as practiced in organic cultivation, affects the growth of toxigenic fungi and whether it carries risks related to mycotoxins. The increasing demand for organic food in developed countries is increasing due to consumer awareness. Cancer, allergies of unknown origin, and fertility issues are increasingly attributed to the contamination of crops with pesticides and industrial chemicals in the air and water [6,7]. This growing demand for organic food is reflected in the expanding acreage dedicated to organic cultivation. Data shows that over the past two decades (2001–2021), the total area under organic cultivation and land undergoing conversion has increased fivefold, reaching 1.3 million hectares. In a single year, from 2020 to 2021, there was a global increase of 1.7% in this area. The organic grape area doubled between 2010 and 2020 and continues to increase every year [8].

However, in unfavorable weather conditions, fruit in organic cultivation is susceptible to mold growth and potential mycotoxin contamination [9]. The sulfur and copper compounds allowed in organic cultivation may not always provide sufficient protection. Therefore, in conventionally grown fruit, it is pesticide residues, whereas in organic farming, it is mycotoxins that may (but not necessarily) pose a risk.

Another critical aspect concerns the nutritional value and health-promoting properties of organically and conventionally grown fruits. Particular attention is given to polyphenolic compounds, which are considered to be known for their health-promoting qualities as antioxidants with positive effects on the human body [10]. These compounds exhibit well-documented antioxidant, antimutagenic [11,12], cardioprotective, anticancer, antiaging, antimicrobial, and anti-inflammatory properties [13,14]. Furthermore, these compounds have an impact on the taste and aroma qualities of both the raw material and the final product, such as wine [15]. The composition of polyphenolic compounds in fruit particularly depends on grape cultivars, cultural and agronomic practices [16,17], and climate conditions [18]. Grape maturity is also a very important parameter, as ripening leads to both quantitative and qualitative modifications in these compounds [19]. The most important polyphenols identified in grapes are anthocyanins, flavanols (also called flavan-3-ols), flavonols, and phenolic acids [20,21]. Dark-skinned fruits, in particular, are rich in these compounds, especially anthocyanins. Moreover, the cultivation method can also influence the technological parameters of the fruit, including the content of basic substances that influence wine taste and the compounds responsible for the proper fermentation of juice [22]. Organic grapes are characterized by higher total acidity, lower pH, a higher content of total polyphenols, including anthocyanins, phenolic acids, and flavonols, and lower volatile acidity compared to conventionally produced grapes [23,24]. Frequently, organically grown fruits are smaller in size compared to conventionally grown ones, which results in a concentration of compounds within the fruit, often increased by unfavorable growing conditions. Such compounds include polyphenols, which influence the plant’s defense response. Additionally, the organic farming system necessitates the use of organic fertilizers, which have a beneficial effect on soil structure by increasing humus and nitrogen content [25]. Consequently, such fruits contain higher nitrogen content (higher yeast assimilable nitrogen—YAN), which significantly influences the fermentation process by which yeast nutrient doses can be reduced.

Therefore, the aim of the research is to obtain information about the impact of plant protection methods and farming practices (organic and conventional) on the quality of fruit, grape juice, and wine. An important aspect of this research is to disseminate and publicize knowledge about the risks and benefits associated with consumption of fruits and fruit preserves grown using these distinct approaches.

## 2. Materials and Methods

### 2.1. Characteristics of the Area of Research and Plant Material

*Vitis vinifera* L. ‘Solaris’ was used for the study (Figure 1). Fruit was taken from an organic and conventional plantation located near Szczecin in north-western Poland (53°21′09.8″ N 14°26′23.3″ E). In the area of Szczecin and in the nearby northern region, minimal temperatures range from −12 °C to −15 °C, which corresponds to values typical of zone 7B. The average temperature during the growing season (April–October) between 1951 and 2019 was 14.3 °C, and rainfall was approximately 350 mm [26].

During 2020–2022, major changes in the weather were observed. In July and August, temperatures exceeded 30 °C for several days and did not fall below 20 °C at night—a phenomenon of tropical nights. There were prolonged periods of drought—2020: 35 days without rainfall; 2021—32 days without rainfall; 2022—46 days without rainfall. This period was followed by heavy/intense rainfall exceeding 100 mm per day. September was characterized by weather typical of the period and similar to that of many years (data obtained from the Meteorological Experimental Station in Lipnik, Poland, 53°20′35″ N, 14°58′10″ E).

### 2.2. Cultivation Scheme

Plants were planted in 2016 at a spacing of 1.01 × 2.28 m (Figure 2). Pruning is carried out in January–February. The plants were pruned with a Guyot (one arm) training system and vertically positioned with eight shoots with two clusters per each. On both plantations in the rows, weeds are removed with a mechanical weeder. In the inter-rows, mixtures are used to improve the soil structure and enrich it with organic matter (lucerne, red clover, oil radish). During the growing season, shoots and excess leaves from the cluster zone are removed mechanically.

During the growing season (May–October), the following was applied for crop protection (in pure component per hectare):

In the organic plantation:sulfur—12.5 kg/ha (withdrawal period—56 days)copper (copper oxychloride and copper hydroxide)—1.75 kg/ha (withdrawal period—7 days)potassium carbonate—17.5 kg/ha (withdrawal period—0 days)potassium gray soap—4 kg/ha (withdrawal period—0 days)

In a conventional plantation:metalaxyl-M (a compound of the phenylacetamide group 3.8%) and mancozeb (a compound of the dithiocarbamate group 64%)—2.25 kg/ha; (withdrawal period—56 days)cyflufenamid (a compound of the phenylacetamide group 5.32%)—0.3 kg/ha; (withdrawal period—21 days)cyprodinil (a compound of the anilinopyrimidine group 37.5%) and fludioxonil (a compound of the phenylpyrrole group 25%)—1.2 kg/ha (withdrawal period—21 days).

Bulk samples from the organic and conventional vineyards (5 kg containers of fruit each) were collected at the end of September/beginning of October 2022 (during the main harvest) in sterile containers. Each bulk sample was taken from 10 grapevines from three representative locations in different parts of the vineyard. After being transported to the laboratory, the juice was extracted from the fruit, and analyses were conducted. Mycotoxins were also determined in the pomace. Additionally, the copper and sulfur content in the wine was analyzed (results are being prepared for publication).

### 2.3. Bacteria and Fungi Microbiome Analysis

A total of 1 g of each sample (grape juice) was ground and homogenized using TissueLyser LT (Qiagen, Düsseldorf, Germany). DNA was isolated from prepared material with a QIAampPowerFecal DNA Kit (Qiagen, Düsseldorf, Germany), and concentrations for each sample were quantified by fluorometric quantitation using a Quantus™ Fluorometer (Promega, Walldorf, Germany).

The gene fragments were amplified with the PCR primers recommended for the Illumina technique. Primers ITS3F (GCATCGATGAAGAACGCAGC) and ITS4R (TCCTCCGCTTATTGATATGC) for fungal ITS library. 341F (CCTACGGGNGGCWG_CAG) and 805R (GACTACHVGGGTATCTAATCC) for bacterial 16S rRNA libraries were employed. The 16S rRNA and ITS gene fragments were amplified with the PCR primers recommended for the Illumina method. The primers were designed by adding Illumina adapter overhang nucleotide sequences to the PCR. Amplicons were indexed using a Nextera^®^ XT Index Kit according to the manufacturer’s instructions. DNA was sequenced in Illumina MiSeq (Illumina, San Diego, CA, USA) in 2 × 250 paired-end mode. The bacterial and fungal NGS sequences of stigmas were submitted to the NCBI Short Reads Archive (SRA) under the project number PRJNA1138628. Each file underwent quality control (QC), which included quality filtering (removing sequences with ≥5 ambiguous base pairs) and length filtering (removing sequences with a length ≥2 standard deviations from the mean).

### 2.4. Juice Quality

The total Soluble Solid Content-SSC (°Bx) in samples was measured at 20 °C by digital refractometer (PAL-1, Atago, Tokyo Japan). Acidity was determined by titration of aqueous extract with 0.1 N sodium hydroxide to an end point with pH 8.1 (Elmetron CX-732, Zabrze, Poland), according to the PN-90/A-75101/04 [27] standard.

The juice’s turbidity was measured using a Lovibond TB211IR working on the principle of measuring scattered light in the 400–600 nm range.

YAN was quantified using the enzymatic method; readings were taken in an automatic wine analyzer using the spectrophotometric method.

The DPPH (1,1-diphenyl-2-picrylhydrazyl) assay was conducted according to the method of Yen and Chen [28]. The FRAP (ferric-reducing antioxidant power) assay was conducted according to the method of Benzie and Strain [29]. For the ABTS+ (2,2′-azobis(3-ethylbenzothiazoline-6-sulfonate)) assay, the procedure was conducted according to a method described previously. The ORAC-FL assay involves measuring the antioxidant capacity of a sample by monitoring the fluorescence decay of fluorescein in the presence of the peroxyl radical generator AAPH. The antioxidant capacity is expressed as millimoles of Trolox per 100 g DW. Measurements in the DPPH and FRAP assays used a UV-2401 PC spectrophotometer.

Polyphenolic compounds were analyzed using the UPLC-PDA-MS/MS Waters ACQUITY system (Waters, Milford, MA, USA) consisting of a binary pump manager, sample manager, column manager, photodiode array (PDA) detector, and quadrupole mass spectrometer with electrospray ionization (ESI) [30].

All tests were performed in three replications.

### 2.5. Mycotoxin Detection

#### 2.5.1. Detection of Mycotoxins in Grape Juice and Pomace

Trichothecenes and Zearalenone Analysis. A 12.5 g sample was mixed with 50 mL acetonitrile/water (80:20, *v*/*v*) for 1 h, then centrifuged at 5000 rpm for 10 min. The supernatant was subjected to a clean-up process using a Bond Elut^®^ Mycotoxin column (Agilent, Santa Clara, CA, USA). An aliquot of 40 µL internal standard solution (13C-ZAN; c = 1000 µg/L) was added to 4 mL of the extract, and then 2 mL of the purified extract was combined with 50 µL internal standards solution (13C-DON; c = 2500 µg/L; 13C-T2; c = 250 µg/L; and 13C-HT2; c = 250 µg/L). This mixture was evaporated to dryness using nitrogen at 45 °C. Subsequently, 495 µL of methanol/water (1:4) was added, and the sample was reconstituted. Mycotoxins were determined using HPLC with MS/MS detection on a Shimadzu Nexera coupled to an API4000 mass spectrometer equipped with a Gemini-NX C18 chromatographic column, employing a gradient elution using 1% acetic acid in water (mobile phase A) and methanol (mobile phase B) with the addition of 5 mM ammonium acetate to both mobile phases.

Patulin Analysis. A 5 g sample was mixed with 20 mL acetonitrile/water (80:20, *v*/*v*) for 1 h, followed by centrifugation at 5000 rpm for 10 min. The supernatant was cleaned up using a push-through-type SPE column, MycoSep 228 AflaPat (Romer, Getzersdorf Austria). The purified eluate (4 mL) was combined with 20 µL of isotopic 13C-labeled PAT (13C-PAT; c = 50 µg/L) and evaporated to dryness under a gentle stream of nitrogen at 45 °C. The residue was then redissolved in 1 mL of a mobile phase mixture, methanol/water (3:7, *v*/*v*), and filtered prior to analysis. Detection was carried out using HPLC Nexera coupled with a 5500 Qtrap mass detector and a Gemini C18 column for separation.

Fumonisins Analysis. A 25 g sample was homogenized with 100 mL acetonitrile/water (50:50) for 3 min. The extract was filtered, and the pH was adjusted to 6–9. Three milliliters of the filtrate were mixed with 8 mL of methanol/water (75:25), and this mixture was applied to a conditioned MultiSep 211 Fum column (Romer Labs, Tulln, Austria). The column was washed with 8 mL of methanol/water (75:25) and 3 mL of methanol. Toxins were eluted using 10 mL of methanol/acetic acid (99:1). The eluate was collected and evaporated to dryness with nitrogen. Then, 1 mL of acetonitrile/water 1:1 solution was added to the vial, and the sample was mixed. Detection was achieved using HPLC Nexera, an API4000 mass spectrometer, and a Gemini-NX C18 column.

Ochratoxin A Analysis. A sample portion (12.5 g) was homogenized with 50 mL acetonitrile/water (60:40) for 2 min. After filtration, a 5 mL aliquot of supernatant was added to a 55 mL PBS solution, and the mixture was filtered again. A total of 48 mL of the diluted extract was applied to an Ochraprep column (Rhone Diagnostic, Glasgow, UK). The column was washed with 20 mL of water and dried with air. OTA was eluted using 1.5 mL of methanol/acetic acid (98:2). The eluate was collected, and 1.5 mL of water was passed through the column, followed by mixing of the sample. Detection was performed using HPLC with fluorescence detection.

Aflatoxins Analysis. To 25 g of the sample, 2.5 g of sodium chloride was added and homogenized with 50 mL of methanol/water (80:20, *v*/*v*) for 1 min. After filtration, 10 mL of the extract was added to 40 mL of water, shaken, and filtered again. A total of 10 mL of the diluted extract was applied to an AflaTest column (Vicam, Watertown, MA, USA). The column was washed twice with 10 mL of water. Aflatoxins were eluted using 1 mL of methanol. The eluate was collected, and 1 mL of water was added before mixing the sample. Aflatoxins were determined using HPLC with FLD, preceded by post-column derivatization.

#### 2.5.2. Detection of the Presence of Genes Encoding Ochratoxins, Aflatoxins, and Patulin in Juice

To prepare the sample for the isolation of total genetic material, 10 mL of the test material in liquid form was concentrated by centrifugation (7000 rcf for 20 min), and the supernatant was decanted. The pellet, which consisted of the concentrated material, was then homogenized in 2 mL of sterile deionized water. For DNA isolation, 250 µL of the homogenate was collected, and the remaining material was dried (using a drying scale—Radwag, Poland) to determine the dry weight. DNA isolation was performed by lysing the collected material in tubes containing a lysis buffer and glass beads. Lysis was carried out for 10 min at a frequency of 50 oscillations per second (using a TissueLyser LT from Qiagen). Subsequently, total DNA was isolated using the Food-Extract DNA Purification Kit (EURx) according to the manufacturer’s instructions. The isolated DNA was used for qPCR reactions to quantitatively determine mycotoxin-producing fungi.

Standards for qPCR reactions were prepared by incorporating amplification products (OTA—*Penicillium verrucosum*, AFLA, and patulin—a sample of degraded food) into a plasmid using the TOPO™ TA Cloning™ Kit, with pCR™ 2.1-TOPO™ (Thermo Fisher Scientific/Invitrogen, Waltham, MA, USA). The copy number of genes was calculated using a calculator available at the DNA calculator.

qPCR reactions for patulin and AFLA were performed following the methodology described by Rodrigez and others [31]. AFLA (GenBank accession number for *omt-1* gene: L25835.1): F-omt GGCCGCCGCTTTGATCTAGG, R-omt ACCACGACCGCCGCC, OMTprobe [HEX]-CCACTGGTAGAGGAGATGT-[BHQ1].

Patulin (GenBank accession number for *idh* gene: AF006680): F-idhtrb GGCATCCATCATCGT, R-idhtrb CTGTTCCTCCACCCA, IDHprobe [FAM]-CCGAAGGGCATCCG_[TAMRA]. Ochratoxin (GenBank accession number for *otanps*PN gene: AY557343): F-npstr GCCGCCCTCTGTCATTCCAAG, R-npstr GCCATCTCCAAACTCAAGCGTG 5185 NPSprobe [Cy5]-CGGCCGACCTCGGGAGAGA[BHQ2].

The reaction mixtures were prepared using a master mix buffer (Maxima Probe 2X, ThermoFisher Scientific), 720 nM of each F-idhtrb and R-idhtrb primer and IDHprobe, 480 nM of each F-npstr and R-npstr primer and 600 nM of NPSprobe, and 80 nM of F-omt primer and 160 nM of each R-omt primer and OMTprobe. Each reaction included 1 μL of DNA template from each of the mycotoxin-producing strains (3 μL of total DNA template) in a final volume of 25 μL. The reactions were carried out using the RotorGene Q platform (Qiagen).

The qPCR program was started with initial incubation at 50 °C for 2 min to activate the uracil-N-glycosylase (UNG) enzyme, followed by an incubation at 95 °C for 10 min to denature the UNG enzyme. The cycling phase included 40 cycles at 95 °C for 30 s and 58 °C for 2 min.

### 2.6. Detection of Pesticide Residues

The analysis was carried out on juice samples prepared from grapes. Each bulk sample was analyzed for the detection of 280 active substances and their metabolites most commonly used in horticultural crops. Gas-tandem chromatography-mass spectrometry was used to detect pesticide-active substances. Analyses were performed in triplicate. This set of tests is performed as a standard for the control of organically grown fruit at The National Institute of Horticultural Research [32]. The contents of copper and sulfur were determined after mineralization in nitric acid and perchloric acid at a ratio of plants of 3:1 [33]. The copper and sulfur contents were measured with flame atomic absorption spectroscopy (iCE 3000 Series).

### 2.7. Statistical Analysis

Statistical analyses were performed with Statistica 12.5 (StatSoft Polska, Cracow, Poland). The Student’s *t*-test was used to compare two means. Fisher’s exact test was used to test statistical significance in contingency table analysis. Statistical methods of data analysis are provided in the descriptions or titles of graphs and tables. The dominance structure was determined based on the following scale: eudominants above 10.0% of all individuals in the compared taxonomic group, dominants: 5.1–10.0%, subdominants: 2.1–5.0%, recedents: 1.1–2.0%, subrecedents below 1.0%. The ecological indicators describing the relative size of a fungal community, dominance, diversity, and evenness were calculated with the use of standard formulas. Principal component analysis (PCA) was performed and visualized in XLSTAT software (2023.2.0.v.1411) [34]. Network analysis was performed using the AxisForce 2 algorithm with a LinLog mode-based Spearman correlation matrix for edges and a cumulative percentage of each variable for nodes (Gephi 0.9).

## 3. Results

### 3.1. Bacterial and Fungal Microbiota of Juice

#### 3.1.1. Bacteria

Different methods of grape cultivation (conventional and organic) influenced the composition of the grape bacterial profile, particularly in terms of the quantitative representation of identified taxa (Table 1). In most cases, similar types of bacteria were detected in both conventionally and organically grown fruits; however, their respective quantitative contributions varied depending on the cultivation method. It is worth noting the disparity in the number of reads for all the identified bacterial sequences (Table 1).

In the case of bacteria originating from organic cultivation, nearly three times more sequences were observed than those detected on conventionally grown fruits. Nineteen additional bacterial taxa were found on organic fruits (Table 1). Bacteria of the genus *Sphingomonas* were the most abundant, regardless of the cultivation method. These dominant bacteria constituted approximately 22.5% of the total identified prokaryotic population. The next most prevalent group of bacteria in both cases (fruits from organic and conventional fields) belonged to the genus *Massilia* (15.7% and 18%, respectively). Notably, there was a statistically significant increase in the presence of these bacteria on fruits treated with conventional pesticides. These two genera of bacteria clearly dominated, regardless of the cultivation method, impacting similar ecological parameters, such as dominance, diversity, and evenness, with respect to the bacterial populations in both cultivation types (Table 1). Bacteria classified into other genera represented a significantly lower percentage. Furthermore, they accounted for significantly fewer representative taxa (ranging from 1.7% to 6.1% of the entire bacterial pool). For instance, *Hymenobacter* was more abundant in fruits from fields protected by conventional pesticides. In contrast, bacteria from the genera *Pseudomonas*, *Variovorax*, *Brevundimonas*, and *Pedobacter* were more abundant in fruits from organic cultivation. The remaining groups were even less represented in the overall bacterial pool (Table 2). In six instances, there was a lack of trace amounts of bacterial representatives from a particular taxon on fruits from one field. This observation mainly concerned less representative taxa on the grape clusters, constituting less than 1% of the total bacterial population. With one exception, the genus *Gluconobacter* is a fairly densely populated fruit from conventional cultivation (5.4% of all bacteria). In organic cultivation, the presence of these bacteria was noted at 0.2%. Similarly, bacteria from the genera *Erwinia* and *Serratia* were present in conventionally grown fruits, but their quantities were low, approximately 1.6% (*Serratia*) and 0.8% (*Erwinia*), while they were either not detected or found in very minimal amounts on organic fruits (0.2% for *Serratia*).

#### 3.1.2. Fungi

Considering the total number of sequence reads (16,390 and 17,125) and the number of differing taxa (33 and 35 OTUs), it can be inferred that the farming system did not exert a significant influence on the qualitative and quantitative composition of grape-inhabiting fungi. However, for a more comprehensive analysis of the research findings, differences in ecological parameters should be specified (Table 1). It is worth noting that the dominance of several taxa was more pronounced in the case of grapes from conventional cultivation. At the same time, greater diversity and evenness within fungal taxa were observed in organic cultivation (Table 1). The most dominant fungi, inhabiting the fruits regardless of the farming system, were *Erysiphe* and *Aureobasidium* genera. The difference lay in the fact that fungi belonging to the *Erysiphe* genus constituted nearly 40% of all identified fungal taxa on fruits from conventional cultivation, which is approximately 18% more than on organic fruits. In the case of fungi from the genus *Aureobasidium*, they were abundant in both organic and conventionally grown fruits (accounting for 29% of all fungi). The subsequent most abundant taxa, irrespective of the type of cultivation, included genera such as *Alternaria*, *Mycosphaerella*, *Botryotinia*, *Cladosporium, Dissoconium*, and *Penicillium*. However, the genera *Botryotinia*, *Mucor*, *Hanseniaspora*, *Mycosphaerella*, and *Dissoconium* were significantly more abundant in organic fruits than in conventionally protected ones. *Hanseniaspora* and *Mucor* were virtually absent from fruits protected with synthetic pesticides. On the other hand, the genus *Penicillium* was less abundant by approximately 50% in organic fruits (Table 3).

Figure 3 presents the co-occurrence analysis of microorganisms, including bacteria and fungi, which form two distinct groups with high internal correlation (Group I and Group II), connected by specific OTUs (Operational Taxonomic Units).

Group I included 14 bacterial OTUs and 9 fungal OTUs. In this group, potentially undesirable fungi such as *Erysiphe*, *Cladosporium*, *Epicoccum* genera, and the potentially antagonistic fungus *Aureobasidium* genus were identified, while the dominant bacteria were *Massilia* and *Sphingomonas* genera.

Group II consisted of 18 bacterial OTUs and 20 fungal OTUs. This group did not have any OTUs with clearly high abundance, but a significant presence of fungi such as *Alternaria*, *Botryotinia*, and *Mycosphaerella* genera was noted, along with bacteria from the genera *Pseudomonas* and *Variovorax*. Both groups were connected by the OTUs *Frondihabitans australicus* and *Rathayibacter* genus, which appeared in both groups, forming a bridge between them. In Group I, the fungus *Aureobasidium* spp. was strongly correlated with several OTUs, as well as with *Botrytis* spp., which belongs to Group II.

It is important to note that these co-occurrence analysis results partially overlap with the biodiversity indices. Moreover, they indicate which OTUs are responsible for biodiversity and dominance depending on the cultivation method, which is particularly evident in the case of the mycobiome. Similar dominance indices for bacteria in both cultivation methods, but a clear overall dominance of *Massilia* spp. and *Sphingomonas* spp., suggest a universal symbiotic relationship between these bacteria and the plant, as well as their resilience to varying biotic and abiotic conditions.

Limited correlations were observed between the OTUs of Group I and Group II, suggesting that the microorganisms within these groups may have different ecological roles and development strategies, both in mutualistic and antagonistic relationships. These differences may affect processes occurring in the environment, such as organic matter degradation, nutrient cycling, or pathogen resistance.

### 3.2. Health Safety-Pesticides and Mycotoxins in Juice

#### 3.2.1. Pesticide Active Ingredients

In organic grape production, sulfur and copper used in vineyard management come from specific permitted organic inputs (Table 4). The organic crop exhibited higher levels of copper in both juice and wine compared to the conventional crop. The organic juice and wine had higher concentrations of copper and sulfur than their conventional counterparts. Specifically, the organic juice and wine had significantly higher levels of sulfur. Importantly, no unauthorized pesticides were detected in the juice from organic fruit. However, high levels of synthetic pesticides were found in two out of three samples from the conventionally cultivated field, with cyprodinil and fludioxonil exceeding the legal limit by more than four times.

#### 3.2.2. Mycotoxins

No mycotoxins (aflatoxins, trichothecenes, zearalenone, patulin, fumonisins, ochratoxin A) were detected in any of the juice and grape marc.

As per the Mann–Whitney test, it was noted that the presence of fungi with the capability to produce aflatoxins and ochratoxin was higher in the case of organic cultivation. However, these differences were not statistically significant (Figure 4). This lack of significance was due to variations in results observed within the field. In the case of results from the ecological field, the parameter value was influenced by the sampling point.

### 3.3. Chemical and Nutritional Properties

The results revealed significant differences in polyphenolic compounds and antioxidant activity between the two cultivation methods (Table 5). Organic cultivation consistently exhibited higher levels of polyphenolic compounds, including hydroxycinnamic acids (such as GRP, caftaric acid, coutaric acid, fertaric acid) and flavan-3-ols (such as procyanidin dimer B1, (+)-catechin, procyanidin dimer B2, and (−)-epicatechin), compared to conventional methods. The predominant compound in the juices was caftaric acid, constituting 61% of all acids, and (−)-epicatechins, making up 75% of all flavan-3-ols. The total content of flavan-3-ols, including various monomers, oligomers, and polymers, was significantly higher in organic juice. The highest increase was in (+)-catechin, which rose by approximately 32% with organic cultivation. Caftaric acid showed a 15% increase, while (−)-epicatechin varied by only 9% between cultivation methods. Gallic acid content was not significantly impacted by the cultivation system.

Additionally, organic cultivation demonstrated superior antiradical activity and reducing power, as indicated by the DPPH, FRAP, ABTS, and ORAC assays (Figure 5). If a lower value indicates greater antioxidant capacity, the interpretation of the results from the graph for grape juice is as follows: Grape juice from organic farming shows greater antioxidant capacity (56.4) compared to juice from conventional farming (62.5). This means that organic juice has a higher ability to neutralize DPPH free radicals. Similarly, in the FRAP test, juice from organic farming demonstrates greater antioxidant activity (78.5) than juice from conventional farming (87.3). This suggests that the organic juice has a greater ability to reduce iron ions (Fe^3+^ to Fe^2+^), which is a measure of antioxidant capacity. In the ABTS test, organic juice again shows higher antioxidant capacity (124) compared to conventional juice (156), meaning that organic juice better neutralizes ABTS+ free radicals. Similarly, in the ORAC test, juice from organic farming has a greater antioxidant capacity (17.3) than conventional juice (22.8). This indicates that organic juice is more effective in neutralizing peroxyl radicals. Grape juice from organic farming shows greater antioxidant capacity compared to juice from conventional farming across all tests conducted. This means that organic juice better neutralizes free radicals and may have a higher health-promoting potential in terms of antioxidant properties.

Based on the presented data regarding the correlations between different phenolic compounds and antioxidant capacity in grape juice from organic and conventional farming, several significant correlations can be observed (Table 6).

In both organic and conventional farming, fertaric acid and gallic acid positively influence the antioxidant capacity measured by DPPH and FRAP, while caftaric acid negatively impacts DPPH. In grapes from organic farming, coutaric acid and (−)-epicatechin were also found to significantly enhance FRAP capacity. In organic farming, gallic acid has a strong positive correlation with ABTS, while fertaric acid shows a strong positive correlation with ORAC. On the other hand, procyanidin dimer B2 and (−)-epicatechin have a negative impact on ORAC results, and in conventional fruits, they also negatively impact the ABTS test.

These findings help identify specific phenolic compounds that contribute to the antioxidant properties of grape samples under different cultivation conditions.

Based on the provided data (Table 7), there is a relationship between the type of crop cultivation (organic or conventional) and several parameters related to crop quality and its processing into wine. The organic crop had a higher level of soluble solids in the juice compared to the conventional crop. This suggests that organic cultivation may contribute to higher sugar content in the grapes, potentially affecting the alcohol content of the resulting wine. The organic crop displayed lower acidity levels compared to the conventional crop. This indicates that organic cultivation may result in lower levels of organic acids in the grapes, impacting the taste and balance of the wine. The conventional crop had higher turbidity levels in the juice, indicating a cloudier juice compared to the clearer juice from the organic crop. The organic crop had higher levels of YAN in the juice compared to the conventional crop. YAN is crucial for yeast growth and fermentation, suggesting that organic cultivation may provide better nutrient availability for yeast during fermentation. Overall, the data indicates that organic cultivation methods can influence several key factors in grape quality and wine production, potentially leading to differences in sugar content, acidity, clarity, and nutrient availability for fermentation.

### 3.4. Interrelationships among Variables

#### 3.4.1. Clustering Analysis

Based on the results of the ANCOVA analysis, it was found that the type of cultivation influenced only a subset of parameters, while the sampling location, including technical replicates, had an impact on seven of them (Figure 6 and Figure 7). However, interactions between the sampling location and variant were observed in eight cases, specifically for GRP, dimer B1, dimer B2, acidity, and NTU. The results indicate a trend of increasing values in the first three parameters (GRP, dimer B1, dimer B2) for the organic cultivation variant. For acidity and NTU, interactions between the sampling location and variant were also observed. This may suggest a rising trend in these parameters for conventional cultivation, although it could also be related to random variability within the field. The analysis results indicate statistically significant differences between the cultivation types for many chemical parameters of grapes. The statistical significance results for individual features are as follows: caftaric acid, coutaric acid II, fertaric acid, (+)-catechin, BRIX, and HAD had very low *p*-values (*p* < 0.001), indicating significant differences between the cultivation types (organic vs. conventional) for these parameters. *p*-values below 0.001 provide strong evidence of statistical significance. For gallic acid and YAN, the *p*-values were lower but still significant (gallic acid: *p* = 0.008, YAN: *p* = 0.048), indicating an impact of the cultivation type on these parameters, although it may be slightly weaker than for the previously mentioned features. For the FRAP parameter, the *p*-value was 0.057, indicating a lack of statistical significance at the *p* = 0.05 level. However, this result suggests the existence of a trend in the differences between the cultivation types for this parameter, although not strong enough to be considered significant. All models had R2 values exceeding 0.5, indicating a strong influence of the main factors and their interactions on the variability of the results. An R2 value between 0.5 and 1 suggests that the model explains a substantial portion of the variability in the data.

#### 3.4.2. Relationships between Variables

Based on the principal component analysis (PCA), it was evident that this analysis accounted for 94% of the variance, indicating that the selected components explain a significant part of the variability in the data (Figure 6). The ordination axis F1, which explained 87% of the result’s variability, proved to be decisive. The PCA results delineate a distinct separation of variants based on the cultivation type. The ecological variants (EV) were closely clustered on the left side of the ordination axis F1, while the conventional variants (CV) formed a tight cluster on the right side of the ordination axis F1. This finding suggests that the cultivation type has a significant influence on the overall variability in the data and that there may be distinct differences in the chemical composition between these two cultivation types. Moreover, there was a high within-object similarity, indicating that the features examined within the same variant exhibited similar values. On the other hand, there was a high dissimilarity between variants, which indicated that the features differed significantly among different variants. These results confirm that there are clear chemical differences between different grape cultivation variants. Based on a detailed analysis of the associations observed among the variables, the following patterns were noticed: for objects with higher values, which are characteristic of the eco variant and demonstrate strong correlations with each other, the following variables were identified: OTA, AFLA, (−)Epicatechin, gallic acid, dimer B1, coutaric acid, dimer B2, coutaric acid II, fertaric acid, GRP, (+)-catechin, microbiome I, cooper, sulfur, BRIX, YAN, and caftaric acid. This means that these variables are related and have similar values for grapes from organic cultivation. On the other hand, for objects with higher values, which are characteristic of the eco variant and are strongly correlated with each other, the following variables were identified: fludioxonil, cyprodinil, microbiome II, NTU, acidity, FRAP, and DPPH. This indicates that these variables are also associated with each other and have similar values for grapes from organic cultivation. In most cases, variables characteristic of the eco variant were negatively correlated with variables characteristic of the conv variant. However, there were exceptions, such as YAN and caftaric acid, which showed a low negative correlation with DPPH and FRAP, and (−)-epicatechin, gallic acid, AFLA, and OTA, which exhibited a low negative correlation with fludioxonil and cyprodinil. It is also worth noting that there was a strong negative correlation between the remaining variables characteristic of the eco variant and those characteristics of the conv variant. This shows that these two groups of variables differed significantly from each other for grapes from different cultivation types.

## 4. Discussion

### 4.1. Influence of the Viticultural System on the Microbiome and Mycotoxin Secretion

Our findings enable a comparison between two cultivation systems. Organic cultivation uses pesticides based on natural active substances such as sulfur and copper, and in conventional cultivation, a number of synthetic pesticides are allowed. In our study, we investigated the differences resulting from the use of the two distinct protection strategies. A significant difference was found regarding the number of reads of all detected bacterial sequences (Reads) on fruit from both types of cultivation. There were almost three times as many bacteria from organically grown fruit. Additionally, there were 19 more bacterial taxa (OTUs) identified on organic fruit. These quantitative differences in bacteria lead to the conclusion that the fungicides used have a substantial impact on the crop. The protection allowed in organic cultivation (pesticides with sulfur, copper, and potassium compounds) favored a more abundant bacterial presence on the fruit compared to synthetic pesticides. However, it should be noted that the ecological parameters assessing diversity and dominance among bacteria did not show significant differences based on the type of cultivation. The cultivation method has a significant effect on bacteria, especially their quantitative representation of the fruit, and indirectly influences their diversity. Previous studies have indicated the negative effect of fungicides on bacteria (reduction in abundance or elimination) [36,37]. Consequently, fungicides may reduce bacterial diversity and the number of bacteria on the fruit, potentially weakening the fruit’s natural biological defense against pathogens.

Such protection may result from antagonistic interactions, such as competition between bacteria and phytopathogenic fungi for habitat [37]. The genus *Sphingomonas* is an aerobic bacteria belonging to the *α*-proteobacteria. It is a highly diverse taxon comprising more than 103 species [38]. Their characteristic feature is the production of an intense yellow pigment [39]. Bacteria of the genus *Sphingomonas* most abundantly colonized grapes regardless of the type of crop. Therefore, it can be concluded that these bacteria react in the same way (or do not show sensitivity) to both natural and synthetic protectants. Information on the *Sphingomonas* resistance to pollutants can be found in the works of the authors of the publication. The ability of *Sphingomonas* strains to biodegrade aromatic hydrocarbons [40,41], ionic liquids such as commercial imidazolium-, pyridinium-, pyrrolidinium-, ammonium-, and phosphonium-based ILs [42], and pesticides [43] has been documented.

These bacteria are widely prevalent in the environment, found in soil and water, and are abundant in plants. They are recognized as part of the plant growth-promoting rhizobacteria (PGPR) and can be located both in the root zone of plants and above the ground [44,45]. Similarly, a bacterium belonging to the genus *Massilia* has been positively identified in the environment [46,47]. These bacteria were notably abundant in grapevine fruit grown in both organic and conventional ways, comprising an average of 16.5% of the total pool of bacteria classified as OTUs. Therefore, it can be concluded that these bacteria react similarly to the pesticides used in both cultivation methods, and they do not lose their beneficial role in promoting plant growth. Additionally, these bacteria have the capacity to parasitize fungi, such as *Pythium*, which helps limit potential yield losses [48]. A high percentage of bacteria from the genera *Sphingomonas* and *Massilia* was recorded in wine after spontaneous fermentation by Chen et al. [49].

In contrast, PGPR from the genera *Variovorax* (including *V. paradoxus*), *Pseudomonas*, *Variovorax*, and *Pedobacter* was found to be sensitive to synthetic pesticides, resulting in a significant reduction in their presence on conventionally protected grapes compared to organically grown grapes. An interesting situation arises in the case of bacteria belonging to the genus *Brevundimonas*, which also displayed sensitivity to pesticides. This bacterium is considered opportunistic, likely due to its isolation from hospital environments [50]. It thus proved difficult to explain the presence of this particular type of bacteria on fruit—almost 4% in organic cultivation and nearly half that amount in conventional cultivation. In 2020, strains of this bacterium were isolated from the soil, specifically from the rhizosphere of potatoes, making it the first description of these strains in soil. The positive role of these bacteria in diazotrophy and the provision of phosphorus to plants was described [51]. Therefore, the detection of this bacterium on grapevines is no longer surprising. Bacteria are not targets of toxic fungicide activity. Nevertheless, the results of the study indicate the coexistence of bacteria in the environment that are both sensitive (described above) and resistant to fungicides [52]. In the current study, some bacteria were also found, which occupied the habitat of the fruit previously inhabited by susceptible microorganisms (bacteria and fungi).

These included species from the genera *Massilia* (included in the PGPR) mentioned earlier, *Hymenobacter* (environmental bacteria, isolated especially from the rhizosphere), *Xylophilus* (bacterial pathogen of grapevines), and *Gluconobacter* (associated with causing fruit spoilage and adulteration of wines) [52,53]. Bacteria of the genus *Xylophilus* were almost twice as abundant in conventionally treated grapes. Bacteria of the genus *Gluconobacter* were less abundant in organic cultivation but clearly present (above 5% OTUs) on fruit that received extensive chemical protection.

Fungicides allowed in both organic and conventional farming had a similar impact on the overall count of fungal sequence reads. The parameter values were similar in both combinations compared (Reads and OTUs). The ecological parameters that describe diversity, evenness in the fungal community, and the dominance index are also worth noting. Based on these data, it can be observed that the fungicides used did not significantly differ in their influence on the number of identified fungi. However, they had a distinct effect on diversity, which was more pronounced compared to the effects seen in the case of bacterial communities (Table 1). Grapes from both crop types were predominantly colonized by fungi belonging to the genus *Erysiphe*, with grapevine powdery mildew attributed to the fungus *Erysiphenecator*. In conventional farming, microorganisms that are sensitive to the pesticide combination used have become extinct or limited. Resistant and also phytopathogenic *Erysiphe* have gained additional habitat to live in, and their numbers have increased. Jones et al. [54] investigated the phenomenon of *Erysiphenecator* acquiring resistance to azole fungicides used intensively in grapevine cultivation. In organic cultivation, where a greater biodiversity of fungi and bacteria was found, *Erysiphe* was about 20% less than on conventionally grown fruit. However, despite the significant contribution of this fungus to the microbiome, no symptoms of powdery mildew were found either on the grapes or on the vine leaves. The fungi thrive especially when sanitary pruning is not performed, the canopy is not thinned, the spring is rainy, and the summer is humid [55,56]. The growing season was dry, so although *Erysiphe* populated the clusters in large numbers, growth did not occur, remaining as spores or at an early stage of development. It is also possible that the abundant fungi of the genus *Aureobasidium* (in our study, they accounted for nearly 30% of the sequence of OTUs in both crop types) limited the proliferation of *Erysiphe* and other identified phytopathogenic fungi. The yeast *Aureobasidium* is described as a beneficial microorganism that stimulates plant immunity, including in grapevines [57]. *Aureobasidium pollutants* occur abundantly on the grapevine, both in the rhizosphere and phyllosphere, including the fruit and consequently also in the juice, affecting the bouquet of natural wine [58]. Some fungal genera were present in lower numbers on conventionally protected (CV) grapes (sensitive to pesticides): *Botryotinia*, *Entylomatales*, *Mycosphaerella* (these are mostly phytopathogenic fungi), and also the genus *Dissoconium* (some beneficial species as they parasitize *Erysiphales, Sclerotinia sclerotiorum*, and other fungi in the phyllosphere) [59]. Some fungi were absent on chemically protected (CV) fruit, although they were present on organic (EV) fruit: *Hanseniaspora* and *Mucor* (the most susceptible). In contrast, fungi of the genus *Penicillium* were almost half as numerous on clusters protected with synthetic fungicides. Regardless of the crop type, the fungi *Aureobasidium*, *Alternaria*, *Cladosporium,* and many others with a negligible percentage of the total microbiome were present in similar numbers.

No mycotoxins were detected in the grape juice or the pomace remaining after pressing. In general, the most commonly detected mycotoxin in grape juices, wines, or grapes is ochratoxin A (OTA). Ochratoxin A is secreted by certain fungal species belonging to the genera *Aspergillus* and *Penicillium* [60]. Fungi of the genus *Aspergillus* were not identified in the tested material, while *Penicillium* fungi were found in quite high numbers. Genes encoding ochratoxin A were also found in the juice of grapes (both organic and those protected with synthetic pesticides). This mycotoxin is produced, among others, by species belonging to *Penicillium*, such as *P. verrucosum* [5]. However, the presence of ochratoxin A itself was not found in the juice or pomace. Although the genes responsible for the trait are present in the genome of the microorganism, their expression does not always occur. Mycotoxins are produced by the mycelium, which grows under favorable conditions during the warm and humid growing season. Without mycelium and suitable weather conditions, mycotoxins are not produced [61]. No fungus was found on the fruit. It is possible that fungi with toxigenic potential remained in spore form. Aflatoxins are produced by fungi of the genus *Aspergillus*, whose presence was not confirmed in the material studied. However, genes encoding aflatoxins were found to be present (Table 6). In view of the fact that only *Aspergillus* produces aflatoxins, the results were puzzling. We suspect that among the taxonomically unidentified fungi (accounting for 1–0.6% in organic and conventional cultivation, respectively) (Table 3), it may have been the fungi of the genus *Aspergillus*, i.e., the ‘owners’ of the genes encoding aflatoxins. It should be clear that there is an increased proportion of genes encoding the tested mycotoxins in the case of organically grown juice (Table 6). So, there is clearly a higher risk of toxin contamination of ‘organic’ juice in the case of favorable conditions for the development of toxigenic fungi. In our experiment, fungi did not develop on the clusters, which prevented the possible production of mycotoxins. Potentially toxigenic *Fusarium* were present in negligible numbers, and therefore, no mycotoxins secreted by these fungi were detected.

### 4.2. Influence of Viticulture Method on the Presence of Pesticide Residues

Copper and sulfur-based fungicides are commonly used in vineyards to control fungal diseases such as grey mold, powdery mildew, and powdery mildew [62]. However, if fungicides are applied in large quantities (due to unfavorable weather conditions and risk of fungal growth) or are not rinsed off by rain, they can remain on the surface and inside the closely packed clusters of the Solaris cultivar for a long time [63,64]. Higher levels of copper and very high levels of sulfur were found in the organic juice compared to the conventional juice. Analysis of the finished wine indicated significantly lower levels of these elements. In contrast, the synthetic pesticides cyprodinil and fludioxonil contaminated juice obtained from conventionally grown grapes. The authors of a report on grape contamination by pesticide residues, Golge and Kabak [65], found exceedances of EU permissible levels in more than 20% of samples from several hundred tested (including high levels of cyprodinil). And nearly 60 percent of the samples contained pesticides in legally permissible amounts. The presence of pesticide residues in the harvested fruit or in the grape juice is often found, even in vineyards following the principles of integrated pest management (IPM) [66]. Some of the active substances, for example, fludioxonil, should be reassessed for safety due to reports of health risks [67]. The presence of pesticide residues in the juice tested may have been influenced by the lack of rainfall during the growing season. Although chemical protection was applied according to guidelines, subsequent doses may have accumulated (Figure 8).

Chemical protection of vines starts in spring, after the start of vegetation, and is carried out throughout the growing season, with withdrawal periods. The measures used were aimed at protecting the plants against powdery mildew and gray mold. These diseases are the biggest problem in grapevine cultivation in our climate. Noticeable, intense climate changes in our region may be associated with more intensive plant protection [68]. Consumption of grapes with high levels of copper can pose a health risk, especially if they are consumed in large quantities or for long periods of time. Copper toxicity can lead to gastrointestinal problems, liver damage, and other health problems [69]. Grapes contaminated with copper can adversely affect wine quality. Copper can react with other compounds during fermentation, leading to unpleasant aromas, shortened shelf life, and stability problems in the finished wine. They can also affect the color, aroma, and flavor profile of the wine [70]. These results imply that the organic cultivation approach potentially led to elevated sulfur content in both the grapes and the resultant wine. Sulfur is one of the few synthetic substances allowed in organic farming due to its relatively low toxicity and compatibility with organic principles [71,72]. Sulfur compounds are a commonly used food preservative. For this reason, consumers may unknowingly ingest high doses of it by consuming many preserved products (for example, dried fruit and fruit juices) in a day. Sensitive individuals suffer from allergies, gastrointestinal problems of varying severity, and asthma [73]. Based on the information provided, it can be concluded that the use of synthetic fungicides had a somewhat limited impact on the typical plant pathogen, *Erysiphe* sp. Despite the targeted protection against this pathogen, synthetic fungicides were not fully effective. However, the fungicide protection was aimed at controlling gray mold (*Botrytis cinerea*) and some saprotrophic fungi that degrade product quality (e.g., *Penicillium* spp.). This contrast in efficacy was less noticeable in organic cultivation. It can be inferred that synthetic fungicides were comparatively more effective in preventing the development of these pathogens in conventionally grown grapes. Nevertheless, despite a well-designed fungicide protection strategy for conventional grapes, it was not possible to completely eliminate the presence of phytopathogens. However, there was a notable reduction in infection levels, which approached economic profitability thresholds (as communicated personally). An important observation is that synthetic fungicides had a nonselective impact on beneficial microbiota, leading to a reduction in nonpathogenic bacterial communities. Furthermore, there was a clear decline in the presence of antagonistic fungi, such as *Hanseniaspora* sp. and *Dissoconium* sp., which exhibited a negative correlation with the occurrence of the *Erysiphe* sp. pathogen. Based on the results from the PCA analysis, certain relationships related to predictors of *Erysiphe* sp. infection severity and other fungal infections become evident. Cooper, sulfur, and (+)-catechin show a strong correlation with the severity of grapevine powdery mildew occurrence. This suggests that higher levels of these components may lead to a greater risk of *Erysiphe* sp. infection. One of the reasons for the higher severity of *Botryotinia* in organic farming compared to conventional farming (Table 3) could be the fungicide protection strategy. Synthetic fungicides used in conventional farming possess different properties, such as surface, translaminar, and systemic actions, which increase their effectiveness in protecting plants against phytopathogens. These fungicides operate both preventively, targeting spores present in the air, and interventionally, targeting infection or advanced disease stages and the pathogen’s development within plant tissues. This approach prolongs the duration of active substances. In the case of copper and sulfur applications, pathogens that have already infected plants can continue to develop after the infection phase. Additionally, the penetration of active substances is confined to the sprayed areas, leaving hard-to-reach places vulnerable to infection, particularly in the case of gray mold (*Botrytis cinerea*).

In terms of co-occurrence correlations, a trend of antagonism between the established groups was observed. For the potential pathogen *Erysiphe* sp., potentially antagonistic taxa like *Massilia* sp. may act antagonistically only towards certain microbial groups, as confirmed by studies on grapevine microbiota by Xu et al. [74]. This suggests that both *Massillia* sp. and potentially antagonistic *Sphingomonas* sp. and *Aureobasidium* spp. may exert antagonistic effects on *Botrytonia* sp., *Mycospharella* sp., and *Alternaria* spp., thereby reducing competition for *Erysiphe* sp. However, as demonstrated by Xu et al. [74], *Massilia* sp. exhibits antagonism against some bacteria inhabiting grapevines, suggesting a mechanism of displacing bacteria antagonistic to *Erysiphe*, such as certain species of *Pseudomonas* spp. and *Variovorax* spp. Despite the higher abundance of bacteria in the organic cultivation variant, they did not significantly reduce pathogens other than *Erysiphe* sp. Furthermore, relatively small changes in the abundance of dominant OTUs indicate the antagonistic or stimulatory effects of a complex of microbial species rather than a single OTU, as well as the involvement of other factors in the restructuring of the microbiota.

### 4.3. Influence of the Cultivation Method of the Vine on the Quality of the Juice—Health-Promoting and Technological Properties

The juice prepared from organic fruit was characterized by better technological parameters. The fruit had a higher SS content and, consequently, more sugars and lower acidity. These parameters directly influence the type of wine and its taste [75]. The juice from organic fruit was also characterized by lower turbidity, allowing the use of lower clarifying agent doses. It also contained a higher amount of nitrogen compounds, which are essential for a proper fermentation process [76]. Grape juices grown under the organic system showed a statistically significantly higher content of polyphenolic compounds compared to the product obtained from conventionally grown fruit. Such observations are also confirmed by other authors [77,78]. The concentration and quality of polyphenolic compounds in plant material are strongly influenced by, among other things, fungicide resistance and the quantity and quality of pesticides used for cultivation [79]. Some herbicides interfere with the synthesis of aromatic amino acids and, consequently, the biosynthesis of polyphenols [79]. Polyphenols present in fruits can inhibit the growth of fungi [80]. Polyphenols have antimicrobial properties that can help prevent the growth of various microorganisms, including fungi. They can interfere with fungal enzymes, disrupt cell membranes, and inhibit fungal spore germination, thereby reducing fungal colonization of the fruit [81,82]. The abundance of polyphenols in fruit from organic plantations may indicate a higher pathogen pressure. Additionally, the microbiome identified on organic fruit also differed significantly from that of conventional fruit. Further research is essential to explore the precise mechanisms underlying the differences in polyphenolic compounds and antioxidant activity between cultivation methods. Furthermore, it is crucial to evaluate the sensory attributes and organoleptic properties of the juice obtained from organic cultivation.

The findings suggest that factors such as fertilization, cultivation practices, and the presence of grapevine powdery mildew can impact the lower sugar content (°Brix) and nitrogen content (YAN) [83,84]. Reduced levels of highly active polyphenols and higher acidity are associated with the severity of secondary fungal infections, aligning with the increased presence of *Penicillium* sp. and *Cladosporium* sp. It is worth noting that a negative correlation exists between microbiota I and the overall antioxidant content (FRAP and DPPH). This suggests that there are other antioxidants, besides the identified polyphenols, that may influence microorganisms, including the pathogenic *Botrytis cinerea* or saprotrophic *Mucor* spp.

The presented results of the correlation between phenolic compounds and antioxidant capacity in grape juice from organic and conventional farming provide significant insights into the impact of specific compounds on antioxidant potential under different cultivation conditions.

The results show that in both organic and conventional farming, fertaric acid and gallic acid play a crucial role in enhancing antioxidant capacity, as observed in the DPPH and FRAP assays. Their positive influence may be related to their well-documented ability to neutralize free radicals and chelate metal ions, which strengthens their protective effect in plant cells [85,86]. These results are consistent with previous studies that demonstrated the strong antioxidant activity of these compounds in other plant products [87,88]. Moreover, gallic acid showed the highest activity in the FRAP assay [89].

On the other hand, caftaric acid shows a negative impact on antioxidant capacity in the DPPH assay. This result may suggest that caftaric acid, despite its phenolic structure, may have a lower ability to neutralize specific free radicals or may act synergistically with other compounds in a way that reduces overall antioxidant capacity. The mechanism of antagonistic interaction may be explained by the formation of adducts and complexes between antioxidants, reduction in a weaker antioxidant by a stronger antioxidant, polymerization of antioxidants, irreversible reactions of radicals leading to their neutralization, and unpredictable reactions between substances [90,91].

In grape samples from organic farming, the role of coutaric acid and (−)-epicatechin is particularly noteworthy, as they significantly increase antioxidant capacity measured in the FRAP assay. The high correlation of these compounds with FRAP may suggest their specific role in reducing iron ions (Fe3+ to Fe2+), which is a key element in assessing antioxidant capacity in this assay [92]. These results may indicate the benefits of organic farming in the context of optimizing the content of these specific phenols in fruits.

Further analyses show that in organic farming, gallic acid has a strong positive correlation with the ABTS assay, while fertaric acid shows a strong positive correlation with ORAC. Procyanidin dimer B2 and (−)-epicatechin have a negative impact on ORAC results, which may be related to their specific interaction with peroxyl radicals generated in this assay. Such behavior of (−)-epicatechin in the ABTS and FRAP assays is confirmed by other studies [93]. Interestingly, these compounds also have a negative impact on ABTS results in fruits from conventional farming, suggesting that their influence on antioxidant capacity may depend on cultivation conditions. Depending on the method used, both synergistic and antagonistic effects were observed in some cases [94].

These results highlight the importance of specific phenolic compounds in shaping the antioxidant capacity of grapes depending on the cultivation method. They also indicate the complexity of interactions between these compounds and different types of redox reactions, which may lead to differences in antioxidant assay results. Further research could contribute to a better understanding of these interactions and to the development of strategies to optimize the content of beneficial phenols in grapes through appropriate management of cultivation methods.

## 5. Conclusions

The viticultural system influenced the microorganisms inhabiting the grapes, with a higher taxonomic diversity observed among microorganisms from organic fruit. Bacterial species belonging to the genus *Sphingomonas* predominated and were equally abundant on fruit from both crops. However, other genera differed in quantity (*Massilia*, *Hymenobacter*, *Pseudomonas*, *Variovorax*, and others). In contrast, fungi displayed a different pattern. The largest number of ITS sequences detected in both types of grapes belonged to phytopathogenic fungi of the genus *Erysiphe*. However, there were definitely more of them on conventionally grown fruit. Less abundant fungal genera, such as *Aureobasidium*, *Alternaria*, *Mycosphaerella*, and *Botryotinia*, were comparable in quantity in both cultivation types. It can be concluded that the presence of numerous antagonistic microorganisms (e.g., species from genera *Sphingomonas*, *Aureobasidium*, *Pseudomonas*, *Variovorax*, etc.) and possibly drought inhibited powdery mildew rather than synthetic pesticides. Synthetic pesticides on the grapes contributed to a lower diversity of microorganisms (OTUs, dominance, diversity). It is possible that phytopathogenic fungi of the genus *Erysiphe* were, therefore, more prevalent on grapes protected with synthetic pesticides. This fact, like the presence of synthetic pesticide residues in the juice, definitely worsened the quality and use parameters of the conventional juice. The organic juice was significantly better in quality. It contained more polyphenols, sugar, and nitrogen, along with lower acidity and turbidity.

High levels of both natural and synthetic pesticide residues in the juice pose a risk, with synthetic pesticides, particularly fludioxonil, being more harmful to health. Sulfur and copper levels in the wine were much lower than in the juice. Sulfites are typically used to preserve wines, making sulfur common in both conventional and organic wines. True powdery mildew infesting the grapes and leaves was not detected, although *Erysiphe* and other molds were found in the composition of the microbiome. This may have been influenced by the weather, low humidity, and lack of rainfall. Therefore, mycotoxins were also not found (a mycelium is needed for the biosynthesis of harmful mycotoxins to occur), although genes encoding aflatoxins and ochratoxins were present in the juice, especially in the organic juice. For organic wines, mycotoxins are a potential threat (although there were no mycotoxins in our wines, there were genes encoding them), while for conventional wines, pesticides are a real threat, as confirmed by the study.

The grape juice from organic farming contains higher levels of total polyphenolic compounds (335.68 mg/L) compared to juice from conventional farming (290.88 mg/L). Specifically, organic grapes have significantly higher concentrations of key polyphenols such as caftaric acid, (+)-catechin, and (−)-epicatechin, which are known for their strong antioxidant properties. The grape juice from organic farming consistently demonstrates superior antioxidant capacity across multiple assays, including DPPH, FRAP, ABTS, and ORAC, compared to juice from conventional farming. Key phenolic compounds such as fertaric acid and Gallic acid were found to play a significant role in enhancing the antioxidant potential, particularly in organic samples. Conversely, caftaric acid showed a negative impact on antioxidant activity, especially in the DPPH assay, suggesting potential antagonistic interactions with other compounds. The results also highlight the complex interplay between different phenolic compounds and their specific effects depending on the method of cultivation, emphasizing the need for tailored farming practices to optimize the health benefits of grape products. Further research is recommended to deepen the understanding of these interactions and to develop strategies for enhancing the phenolic content in grapes through optimized agricultural practices.

In summary, organic juice displayed better quality and usability parameters and was free of mycotoxins. However, it contained sulfur and copper, although their amounts were reduced during the vinification process to comply with standards. Conventional juice had poorer technological parameters and higher pesticide residues, but no mycotoxins were detected. It was found that the presence of mycotoxin genes was higher in organic grapes than in conventional ones. Despite the presence of genes encoding mycotoxins, they were not detected in the wines. Although this was a one-year study conducted on a large group of plants, the findings allow for cautious conclusions. Nevertheless, further research is required to confirm these results and provide a more comprehensive understanding of the impact of farming systems on juice quality and safety.

## Figures and Tables

**Figure 1 antioxidants-13-01214-f001:**
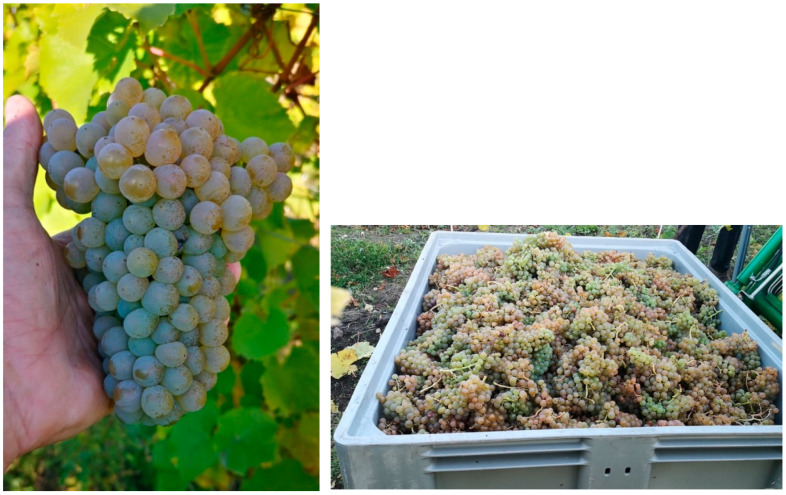
Fruit of the Solaris cultivar on an organic plantation (Fot. I. Ochmian).

**Figure 2 antioxidants-13-01214-f002:**
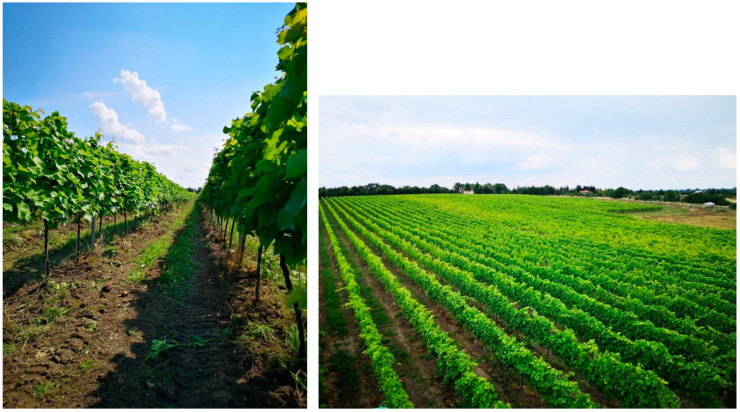
Organic vineyard plantation (Fot. I. Ochmian).

**Figure 3 antioxidants-13-01214-f003:**
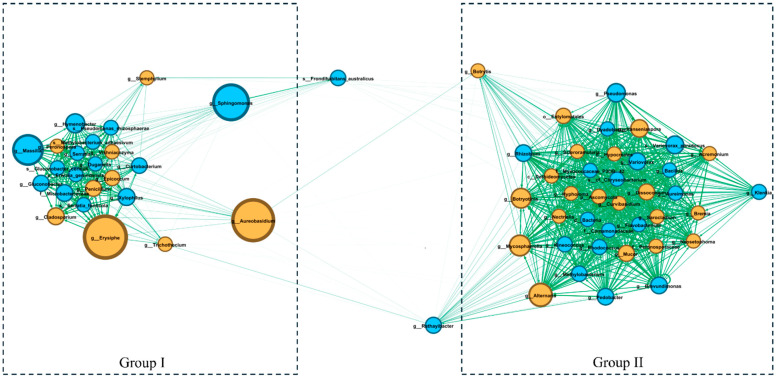
Network co-occurrence analysis between bacteria and fungi OTUs. Nodes (circles) describe bacterial (blue) or fungal (brown) OTUs. The size of nodes symbolizes the feature importance (normalized mean) for the entire sample. Edges (lines) between circles describe the correlation between them. The correlation increases with the intensity of the green color. The correlation increases with the intensity of the green color. Group I and Group II indicate OTUs with a strong intra-group correlation.

**Figure 4 antioxidants-13-01214-f004:**
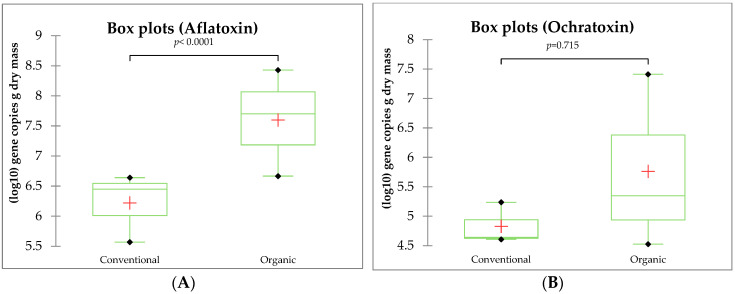
Copies of aflatoxins genes (**A**) and ochratoxin (**B**) (genes g dry weight) in organic and conventional juices. In the boxplot, from the first quartile to the third quartile, mean value, median, maximum, and minimum are marked.

**Figure 5 antioxidants-13-01214-f005:**
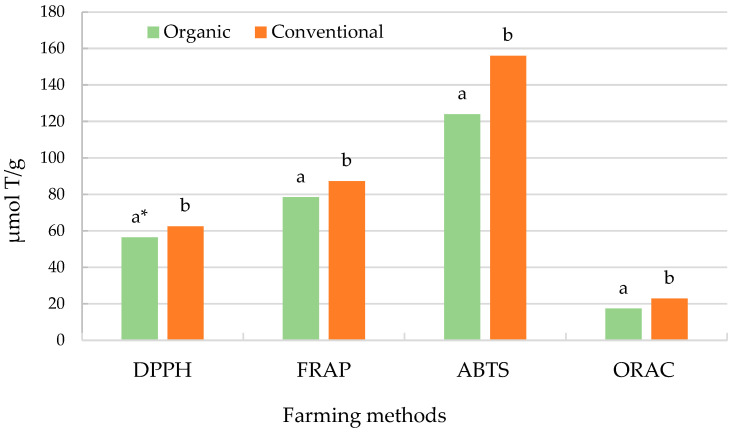
Antioxidant activity in grape juice. * Mean values denoted by the same letter do not differ statistically significantly at 0.05, according to the *t*-test.

**Figure 6 antioxidants-13-01214-f006:**
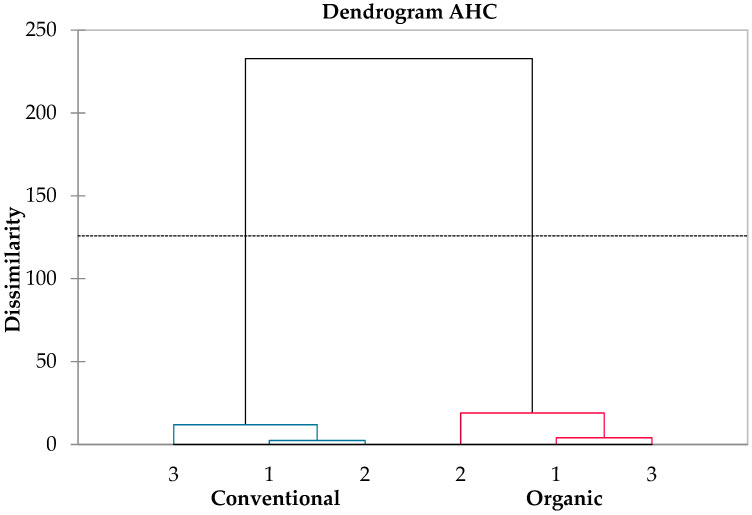
Agglomerative clustering analysis (Ward’s method based on Bray-Curtis dissimilarity matrix) of the distance between differences tested parameters representing organic and conventional juice. Samples were taken from 3 different places in each field (1, 2, 3).

**Figure 7 antioxidants-13-01214-f007:**
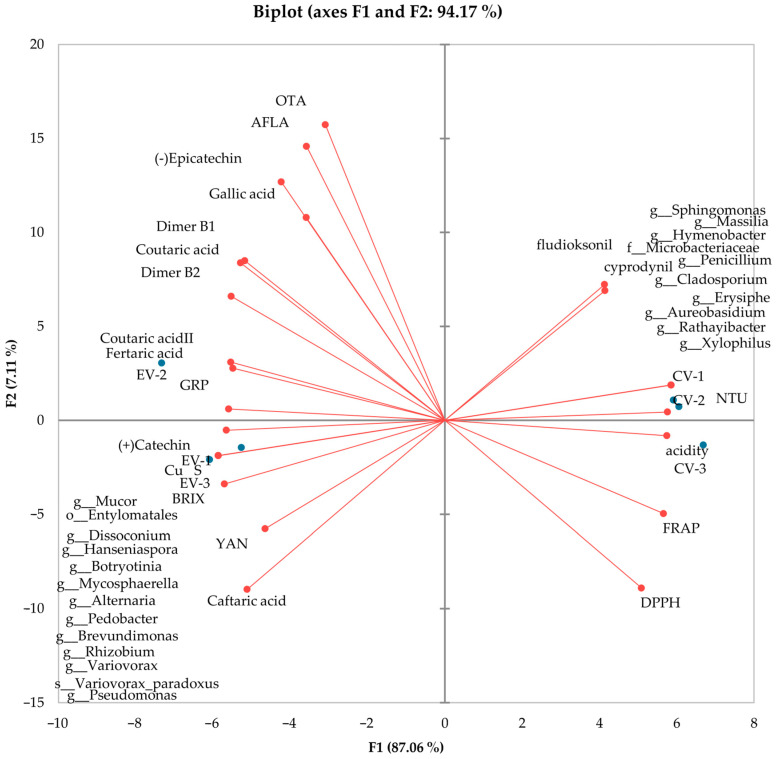
PCA biplot shows the relation between the most abundant orders of bacteria, fungi, and tested parameters and their influence on ecological (EV 1, 2, 3) and conventional samples (CV 1, 2, 3). Blue points show variables (treatments), red lines with dots show vectors of observations (principal components).

**Figure 8 antioxidants-13-01214-f008:**
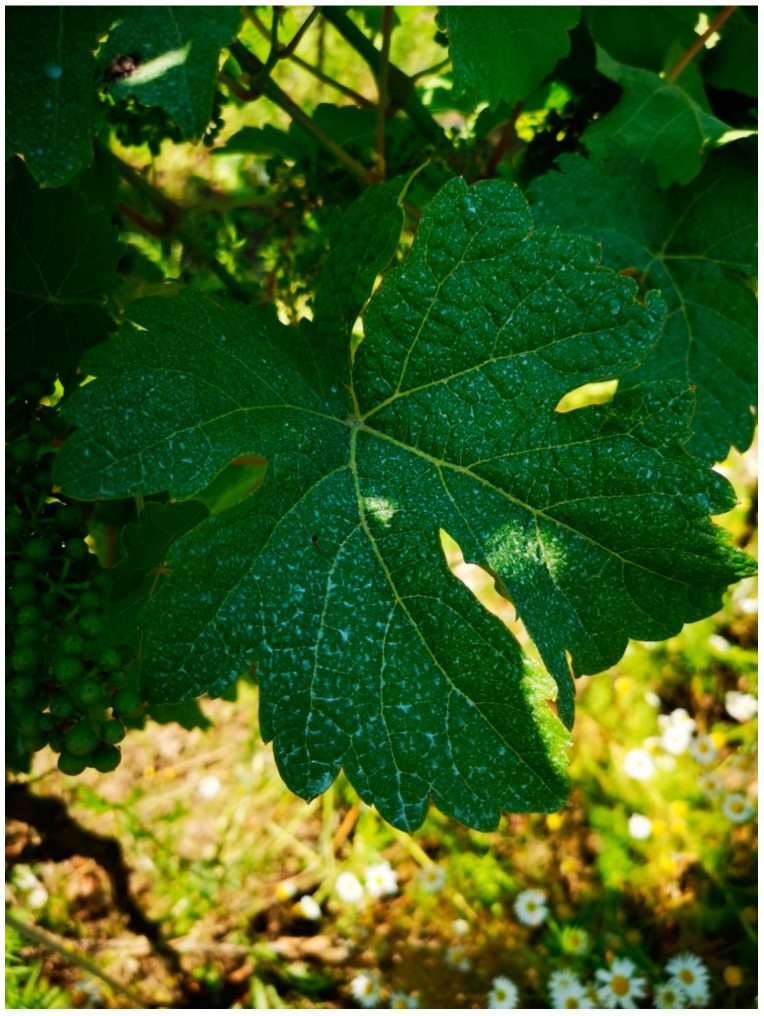
Sulfur and copper solution on grape leaves in an organic plantation (Fot. I. Ochmian).

**Table 1 antioxidants-13-01214-t001:** Taxonomic diversity indicators (based on the spread of 16S rRNA/ITS). OTUs and diversity indices of microbiota identified in organic and conventional grape juice.

	Bacteria (16S rRNA)		Fungi (ITS)	
	Farming Methods			
Index	Organic	Conventional	Organic	Conventional
Reads	26,295	9376	16,390	17,125
OTUs	108	89	35	33
Simpson’s dominance (λ)	0.092	0.099	0.155	0.246
S–W diversity (H’)	3.126	3.080	2.255	1.814
Pielou’s evenness (J’)	0.668	0.686	0.634	0.519

**Table 2 antioxidants-13-01214-t002:** Frequency (%) of bacteria identified in organic and conventional juice. The composition of the bacterial microbiota, OTUs bacteria >0.1% for the tested variants.

Operational Taxonomic Units (OTUs)	Organic Juice	Conventional Juice	Fisher’s Exact Test
*g__Sphingomonas*	22.47	22.61	0.795
*g__Massilia*	15.68	17.96	<0.0001 *
*g__Hymenobacter*	4.84	6.14	<0.0001
*g__Pseudomonas*	5.27	4.19	<0.0001
*s__Variovorax_paradoxus*	4.88	3.57	<0.0001
*g__Variovorax*	3.73	2.85	<0.0001
*g__Variovorax*	3.99	1.77	<0.0001
*g__Brevundimonas*	3.86	1.74	<0.0001
*g__Pedobacter*	3.28	2.01	<0.0001
*f__Microbacteriaceae*	2.80	2.97	0.405
*g__Xylophilus*	2.25	4.04	<0.0001
*g__Rathayibacter*	2.07	2.08	0.966
*g__Methylobacterium*	1.85	1.40	0.004
*g__Curtobacterium*	1.67	1.75	0.641
*s__Frondihabitans_australicus*	1.56	1.57	0.923
*g__Kineococcus*	1.43	1.06	0.005
*g__Klenkia*	1.36	0.95	0.002
*g__Rhodococcus*	1.32	0.82	<0.0001
*g__Gluconobacter*	0.00	4.40	<0.0001
*s__Pseudomonas_rhizosphaerae*	0.65	0.91	0.012
*s__Methylobacterium_adhaesivum*	0.56	0.89	0.001
*s__Gluconobacter_cerinus*	0.19	1.05	<0.0001
*g__Aureimonas*	0.75	0.29	<0.0001
*f__Comamonadaceae*	0.75	0.13	<0.0001
*d__Bacteria*	0.76	0.00	<0.0001
*g__Brevundimonas*	0.62	0.17	<0.0001
*s__cf._Chryseobacterium*	0.64	0.00	<0.0001
*g__Flavobacterium*	0.54	0.12	<0.0001
*f__Myxococcaceae*	0.50	0.22	0.000
*g__Bacillus*	0.47	0.23	0.002
*g__Dyadobacter*	0.49	0.00	<0.0001
*g__Serratia*	0.17	0.82	<0.0001
*g__Duganella*	0.10	0.59	<0.0001
*s__Erwinia_gerundensis*	0.00	0.84	<0.0001
*s__Serratia_fonticola*	0.00	0.80	<0.0001

* Significant *p*-values are in (red font). The intensity of the color symbolizes the affiliation of a given OTU to a dominance class: dark color—eudominant (>10%), moderate color—dominant (5–9.99%), light color—subdominant (2–4.99%), very light color—rare OTUs (1–1.99%), no color—occasional OTUs (<1%).

**Table 3 antioxidants-13-01214-t003:** Frequency (%) of fungi identified in organic and conventional juice. The composition of the fungal microbiota, OTUs fungi > 0.1% for the tested variants.

Operational Taxonomic Units (OTUs)	Organic Juice	Conventional Juice	Fisher’s Exact Test
*g__Erysiphe*	20.32	38.15	0.008 *
*g__Aureobasidium*	28.89	28.98	1.000
*g__Alternaria*	9.57	8.95	0.814
*g__Mycosphaerella*	7.79	6.23	0.593
*g__Botryotinia*	7.15	4.92	0.568
*g__Hanseniaspora*	7.49	0.03	0.003
*g__Cladosporium*	3.01	3.14	1.000
*g__Dissoconium*	3.04	1.37	0.369
*g__Penicillium*	1.26	3.01	0.621
*g__Mucor*	4.34	0.05	0.059
*o__Entylomatales*	2.00	0.84	0.621
*g__Epicoccum*	1.24	1.52	1.000
*p__Ascomycota*	0.96	0.62	1.000
*g__Vishniacozyma*	0.37	0.74	1.000
*g__Neosetophoma*	0.41	0.09	0.497
*g__Stemphylium*	0.21	0.29	1.000
*g__Sarocladium*	0.23	0.16	1.000
*g__Trichothecium*	0.31	0.09	1.000
*o__Hypocreales*	0.23	0.16	1.000
*g__Acremonium*	0.18	0.05	1.000
*g__Peronospora*	0.07	0.13	1.000
*f__Peronosporaceae*	0.12	0.07	1.000
*g__Botrytis*	0.09	0.08	1.000
*c__Dothideomycetes*	0.08	0.06	1.000
*g__Curvibasidium*	0.12	0.04	1.000
*g__Hypholoma*	0.12	0.04	1.000
*g__Scleroramularia*	0.10	0.04	1.000
*g__Bremia*	0.07	0.02	1.000
*g__Nectriella*	0.08	0.00	1.000
*g__Tricladium*	0.05	0.01	1.000
*g__Fusarium*	0.00	0.06	1.000

* Significant *p*-values are in (red font).

**Table 4 antioxidants-13-01214-t004:** Natural and synthetic pesticide residue content.

Methods of Cultivation		Sampling Points	Active Substance (mg/kg)	
			Sulphur	Cooper
			juice	
Organic	natural pesticides	1	17.6	3.26
		2	17.7	3.15
		3	17.0	3.01
		mean	17.4 B *	3.14 B
			wine **	
		mean	6.7 b	0.56 b
Conventional			juice	
		1	3.2	2.14
		2	3.4	2.05
		3	3.1	2.06
		mean	3.2 A	2.08 A
			wine	
		mean	2.8 a	0.44 a
		**Sampling Points**	**Cyprodinil *****	**Fludioxonil**
			juice	
Organic	synthetic pesticides		n.d.	n.d.
Conventional		1	14.7 a *	18.9 a
		2	18.4 a	15.3 a
		3	-	-

* Mean values in columns denoted by the same uppercase or lowercase letters do not differ statistically significantly at 0.05 according to the *t*-test; ** wine data are published elsewhere (in press); *** Limits on the number of pesticides in relation to fresh fruit: cyprodinil 3.0 mg kg, fludioxonil 4.0 mg kg. Regulation (EC) No 396/2005 [35] of the European Parliament and of the Council of 23 February 2005 on maximum residue levels of pesticides in or on food and feed.

**Table 5 antioxidants-13-01214-t005:** Quantification of polyphenolic compounds and antioxidant activity in grape juice.

Polyphenolic Compounds (mg/L)	Farming Methods	
	Organic	Conventional
GRP (*cis*- and *trans*- isomers)	64.68 b *	56.39 a
Caftaric acid (*cis*- and *trans*- isomers)	133.42 b	113.27 a
Coutaric acid (*cis*- and *trans*- isomers)	5.66 b	5.02 a
Coutaric acid (*cis*- and *trans*- isomers)	11.47 b	9.93 a
Fertaric acid	2.76 b	2.36 a
Gallic acid	1.64 a	1.53 a
hydroxycinnamic acids total	219.63 b	188.5 a
Procyanidin dimer B1	13.38 b	11.99 a
(+)-Catechin	14.19 b	9.68 a
Procyanidin dimer B2	3.24 b	2.81 a
(−)-Epicatechin	85.24 b	77.89 a
flavan-3-ols total	116.05 b	102.37 a
Total polyphenolic	335.68 B	290.88 A

* Mean values denoted by the same letter do not differ statistically significantly at 0.05, according to the *t*-test.

**Table 6 antioxidants-13-01214-t006:** Correlation between Polyphenolic Compounds and Antioxidant Capacity of Grape Fruits Depending on the Cultivation Method.

	DPPH	FRAP	ABTS	ORAC
**Farming methods—organic**				
GRP (*cis*- and *trans*- isomers)	0.14	0.22	0.00	0.08
Caftaric acid (*cis*- and *trans*- isomers)	−0.44	−0.04	0.20	−0.21
Coutaric acid (*cis*- and *trans*- isomers)	0.29	0.57	0.23	−0.05
Coutaric acid (*cis*- and *trans*- isomers)	−0.38	0.15	−0.02	−0.30
Fertaric acid	0.45	0.44	0.27	0.48
Gallic acid	0.33	0.35	0.67 *	0.33
Procyanidin dimer B1	−0.06	0.16	−0.04	0.14
(+)-Catechin	0.15	−0.31	−0.47	0.29
Procyanidin dimer B2	−0.01	0.03	−0.11	−0.67
(−)-Epicatechin	0.08	0.64	0.44	−0.30
**Farming methods—conventional**				
GRP (*cis*- and *trans*- isomers)	−0.53	−0.58	−0.19	0.02
Caftaric acid (*cis*- and *trans*- isomers)	0.05	−0.34	−0.06	0.08
Coutaric acid (*cis*- and *trans*- isomers)	−0.08	0.78	0.66	−0.63
Coutaric acid (*cis*- and *trans*- isomers)	0.13	−0.49	−0.04	−0.08
Fertaric acid	0.62	0.80	−0.46	0.36
Gallic acid	0.05	0.83	0.39	0.46
Procyanidin dimer B1	−0.46	−0.40	-0.36	0.66
(+)-Catechin	−0.17	−0.36	-0.31	0.41
Procyanidin dimer B2	0.25	0.07	-0.69	−0.05
(−)-Epicatechin	−0.02	0.47	0.19	−0.79

* Significant *p*-values are in (red font).

**Table 7 antioxidants-13-01214-t007:** Technological parameters of juice.

Parameters	Farming Methods	
	Organic	Conventional
SSA (% Brix)	24.8 b *	22.5 a
Acidity (g/L)	7.5 b	8.6 a
Turbidity of juice (NTU)	1116 a	1388 b
YAN—Yeast Assimilable Nitrogen (mg/L)	147 b	117 a

* See Table 5.

## Data Availability

The data presented in this study are available upon reasonable request from the corresponding author. The samples and any additional research data are available from the authors upon request.

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
