# Peer review of "Antioxidant, Nutritional Properties, Microbiological, and Health Safety of Juice from Organic and Conventional ‘Solaris’ Wine (*Vitis vinifera* L.) Farming"

_antioxidants, 2024, doi:10.3390/antiox13101214_

Round 1

Reviewer 1 Report (Previous Reviewer 1)

The authors have answered all my concerns.

Nothing

Author Response

Szczecin, 02.10.2024

antioxidants-3218506

Response to Reviewers' Comments 

Dear Reviewer,

Thank you for your detailed and constructive feedback on our manuscript Antioxidant, Nutritional Properties, Microbiological and Health Safety of Juice from Organic and Conventional Wine (Vitis vinifera L.) Farming. We have carefully addressed all your comments, as outlined below. All changes have been highlighted in the revised manuscript.

  1. Title Modification:

   We have improved the title by including the cultivar name. The study is a comparative analysis of fruit traits of the same Vitis vinifera cultivar, Solaris, grown under two different farming systems. This suggestion has been implemented in the title.

  1. Keywords Adjustment:

   In the keywords, we have replaced "environmental protection" with "plant protection" to more accurately reflect the chemical agents used for crop protection.

  1. Clarification of Study Design:

   We have indicated that this is a pilot study with a one-year scope, both in the abstract and the conclusion, as requested. Given the one-year experimental plan, we have stressed the need for caution in interpreting the results.

  1. Materials and Methods Update:

   The study was performed using a single collection of plant material. We have added this information in the Materials and Methods section, specifying the time of the year and the month/year of grape sample harvest.

  1. Abstract Rewording:

   The term “physical and functional properties” has been rephrased as “technological parameters,” which are detailed in Table 7. This clarification was added to the abstract (line 24).

  1. Clarification in Cultivation Scheme:

In previous years, the harvest took place around the end of September and the beginning of October, with a possible variation of +/- 10 days. This was taken into account when determining the application time of the respective treatment, considering the withdrawal period. We have added this information to the relevant section.

  1. Clarification in Figure 3:

   We have indicated what the colored circles represent in Figure 3. - Nodes (circles) describe bacterial (blue) or fungal (brown) OTUs. The size of nodes symbolizes the feature importance (normalized mean) for the entire sample. Edges (lines) between circles describe the
correlation between. The correlation increases with the intensity of green color.

  1. Section Rephrase and Sampling Details:

   We have corrected the chapter title and clarified that the samples were collected from three locations, with 10 grapevines per site, resulting in three bulk samples for each cultivation system. This is explained in section 3.4, and we have rephrased the title as " Interrelationships Among Variables."

  1. Weather and Soil Condition Impact:

   Since the plantations are located next to each other and share similar soil conditions, we clarified that the weather and soil variability had no significant effect on the parameters studied. This removes the concern of variable location effects, as detailed in the revised text.

We hope the changes address your concerns and improve the clarity and quality of the manuscript.

Best regards, 

Ireneusz Ochmian and co-autors 

Reviewer 2 Report (New Reviewer)

The manuscript addresses an interesting topic regarding the comparative analysis of the antioxidant and nutrient content and the microbiome profiling of grapefruits from the same cultivar, SOLARIS, grown under organic and conventional farming systems. The experimental plan and methods are clearly presented, describing current chemical analyses and NGS method to determine the impact of the cultivation system on the quality, nutritional value and safety of grape juice samples. The manuscript is well organized and written; however, there are some issues that the authors should clarify and elaborate. These are as follows:

1.       The study is a comparative analysis of fruit traits of the same Vitis vinifera cultivar, SOLARIS, grown under two different farming systems. Thus, suggestion is to depict this in the title as well.

2.       In the keywords the term “environmental protection” should be rephrased to “plant protection” to clearly indicate the chemical agents used for crop protection.

3.       Given that the experimental plan and results refer to a one-year period, lacking replication, and reproducibility, then all results and conclusions should be treated with caution as first indication. This should be indicated also in the Abstract.

4.       Following (3) that the study was performed only once, using a single collection of plant material, then this information should be clearly presented in the Materials and Methods section, indicating the time of the year, month/year, (ln 228) the harvest of grape samples occurred.  

5.       In the Abstract ln 24, what do the terms “physical and functional properties” refer to? Rephrase as these terms are not used in the text.

6.       In section 2.2. Cultivation scheme, the time of each treatment, especially the last application, in both cultivation schemes should be presented in ln 213-226. Also, how many days after the last application, the harvest of samples occurred should be depicted. This information should be discussed in correlation to the regulations for harvest time of grapes, as the study raises health safety issues.

7.       In Figure 3 what the colored circles represent should be indicated.

8.       In section 3.4. “Influence of Sampling Site and Interrelationships Among Variables”, should be rephrased as sampling site and cropping scheme is the same parameter. Usually in agronomic experiments sampling site refers to samples obtain as independent from various sites within the field representing possibly a variability of microenvironments. On the other hand, the study used a single bulk sample obtain from 10 grapevines from each cultivation scheme, type.

9.       As the properties of grapes assessed, such as antioxidants, phenolics, and microbiome, could be affected by weather conditions particularly humidity and precipitation, it would be nice to show whether these climatic parameters affect the grape quality parameters assessed in the two types of cultivation systems.

see above text

Author Response

Szczecin, 02.10.2024

antioxidants-3218506

Response to Reviewers' Comments 

Dear Reviewer,

Thank you for your detailed and constructive feedback on our manuscript Antioxidant, Nutritional Properties, Microbiological and Health Safety of Juice from Organic and Conventional Wine (Vitis vinifera L.) Farming. We have carefully addressed all your comments, as outlined below. All changes have been highlighted in the revised manuscript.

  1. Title Modification:

   We have improved the title by including the cultivar name. The study is a comparative analysis of fruit traits of the same Vitis vinifera cultivar, Solaris, grown under two different farming systems. This suggestion has been implemented in the title.

  1. Keywords Adjustment:

   In the keywords, we have replaced "environmental protection" with "plant protection" to more accurately reflect the chemical agents used for crop protection.

  1. Clarification of Study Design:

   We have indicated that this is a pilot study with a one-year scope, both in the abstract and the conclusion, as requested. Given the one-year experimental plan, we have stressed the need for caution in interpreting the results.

  1. Materials and Methods Update:

   The study was performed using a single collection of plant material. We have added this information in the Materials and Methods section, specifying the time of the year and the month/year of grape sample harvest.

  1. Abstract Rewording:

   The term “physical and functional properties” has been rephrased as “technological parameters,” which are detailed in Table 7. This clarification was added to the abstract (line 24).

  1. Clarification in Cultivation Scheme:

In previous years, the harvest took place around the end of September and the beginning of October, with a possible variation of +/- 10 days. This was taken into account when determining the application time of the respective treatment, considering the withdrawal period. We have added this information to the relevant section.

  1. Clarification in Figure 3:

   We have indicated what the colored circles represent in Figure 3. - Nodes (circles) describe bacterial (blue) or fungal (brown) OTUs. The size of nodes symbolizes the feature importance (normalized mean) for the entire sample. Edges (lines) between circles describe the
correlation between. The correlation increases with the intensity of green color.

  1. Section Rephrase and Sampling Details:

   We have corrected the chapter title and clarified that the samples were collected from three locations, with 10 grapevines per site, resulting in three bulk samples for each cultivation system. This is explained in section 3.4, and we have rephrased the title as " Interrelationships Among Variables."

  1. Weather and Soil Condition Impact:

   Since the plantations are located next to each other and share similar soil conditions, we clarified that the weather and soil variability had no significant effect on the parameters studied. This removes the concern of variable location effects, as detailed in the revised text.

We hope the changes address your concerns and improve the clarity and quality of the manuscript.

Best regards, 

Ireneusz Ochmian and co-autors 

Round 2

Reviewer 2 Report (New Reviewer)

The authors have adequately addressed the comments improving the manuscript.

Two points that need correction are as follows:

1.       Figure 3, what Group I and Group II refer to should be presented in the legend, as well as in the text referenced, ln 424-428.

2.       In the Conclusion section, ln 1024 “and fewer genes encoding them” should be omitted. A separate sentence should be added indicating that the presence of mycotoxins genes was found to be higher in organic grapes than in conventional, based on data shown in Figure 4 and the statement ln 998-999 “For organic wines, mycotoxins are a potential threat 998 (although there were no mycotoxins in our wines, but there were genes encoding them).

The authors have adequately addressed the comments improving the manuscript.

Two points that need correction are as follows:

1.       Figure 3, what Group I and Group II refer to should be presented in the legend, as well as in the text referenced, ln 424-428.

2.       In the Conclusion section, ln 1024 “and fewer genes encoding them” should be omitted. A separate sentence should be added indicating that the presence of mycotoxins genes was found to be higher in organic grapes than in conventional, based on data shown in Figure 4 and the statement ln 998-999 “For organic wines, mycotoxins are a potential threat 998 (although there were no mycotoxins in our wines, but there were genes encoding them).

Author Response

Szczecin, 05.10.2024

antioxidants-3218506

Response to Reviewer Comments 

Dear Reviewer,

Thank you for your constructive feedback and for helping us improve the manuscript. We have addressed both of your suggestions as follows:

  1. Figure 3 - Group I and Group II Clarification:

   As per your recommendation, we have added an explanation of what Group I and Group II refer to, both in the figure legend and in the text, specifically in lines 424-428. These groups now have clear definitions to enhance the reader’s understanding.

  1. Conclusion Section Adjustment:

   We have followed your suggestion and removed the phrase "and fewer genes encoding them" from line 1024. Additionally, we have added a new sentence that highlights the higher presence of mycotoxin genes in organic grapes compared to conventional ones, as shown in Figure 4. This is consistent with the statement in lines 998-999: "For organic wines, mycotoxins are a potential threat (although there were no mycotoxins in our wines, but there were genes encoding them)."

All changes have been highlighted in yellow in the revised manuscript.

We trust these modifications meet your expectations and further improve the clarity and quality of our work. Thank you once again for your valuable comments.

Best regards, 

Ireneusz Ochmian and co-authors

This manuscript is a resubmission of an earlier submission. The following is a list of the peer review reports and author responses from that submission.

Round 1

Reviewer 1 Report

The author just test a lot of data, how does this relate to Antioxidant?

1) The author's Figures and Tables are confusing (2 Figure1 and 2 Figure2), and I don't really understand the purpose of putting the first Figure1, the first Figure2 and Figure3 in the text. Table 6 and the second Figure1 are supposed to present the same result, why are they presented separately?

2) All the chemical formulas and units are not noted for superscript and subscript.

3) In the second Figure2, the text is stacked on top of each other and cannot be read.

4) Line 764, ‘Influence of viticulture method on the presence of pesticide residues’ Bolded? Why?

5) Line 180, ‘One grame’ should be ‘1 g’.

6) Line 130, ‘Vitis vinifera L.’ should be italic.

Author Response

Response to Reviewers' Comments

Dear Reviewer,

Thank you for your detailed and constructive feedback on our manuscript. We have thoroughly addressed all the major and minor comments provided by the reviewers, as outlined below:

Major Comments:

  1. Figures Improvement and Reproducible Script:

   We have enhanced the quality of all figures. The figures are now of higher resolution and clarity. We have also created new figures based on the additional statistical analyses performed.

  1. Statistical Analysis:

We performed additional statistical analyses. The results of these analyses are now included in the revised manuscript along with new, improved figures.

  1. Results Structure:

The results section has been reorganized into three main sections: 3.1. Microbiological Analysis and Mycotoxin Detection, 3.2. Pesticide Residue Analysis and Health Safety and 3.3. Chemical and Nutritional Properties

 This restructuring enhances the logical flow and readability of the findings. Additionally, we have performed further analyses and included new figures to support our findings.

  1. Pictures of Grapes and Cultivation Gardens:

The pictures of grapes and cultivation gardens remain in the main text. All pictures and figures have been uniformly and logically renumbered to ensure consistency throughout the manuscript.

  1. Clarification on "Must" in Tables:

The term "must" has been clarified in the tables and throughout the text. "Must" refers to freshly crushed grape juice that contains the skins, seeds, and stems of the fruit.

Minor Comments:

  1. Integration of Undetected Endophytic Toxins:

The undetected endophytic toxins have been integrated into Table and described in the main text.

  1. Combination of Figures:

We have ensured that all figures and tables are properly labeled and presented without duplication. Table 6 has been removed, and its results are now presented in a newly created figure for better clarity.

Tables 4 and 5 have been combined into a single table to streamline the presentation of data.

  1. Duplicate Figures:

  The duplication of figures has been corrected. There is now only one Figure 2, and the fonts have been adjusted to ensure clarity and readability.

  1. Reorganization of Topics:

The discussion on the effect of viticulture systems on toxigenic fungi and pesticide residues has been reorganized for better clarity and flow.

  1. Line 786 "Fot.3":

  The term "Fot.3" has been replaced with "Fig. 3" for consistency.

  1. Line 130:

 The Latin name of the grape variety has been standardized to "Vitis vinifera L. cultivar Solaris" throughout the text.

  1. Chemical Formulas and Units:

 The chemical formulas and units have been uniformly presented as words, abbreviations, or chemical formulas without mixing. Superscripts and subscripts of Arabic numerals have been corrected where necessary.

Additional Corrections:

  1. Confusing Figures and Tables:

We have ensured that all figures and tables are properly labeled and presented without duplication.

  1. Superscript and Subscript Notation:

 All chemical formulas and units have been corrected for superscript and subscript notation to maintain consistency.

  1. Figure Readability:

The text in the Figure 6 has been adjusted to avoid stacking and ensure readability.

  1. Bolded Section Title:

  The section title "Influence of viticulture method on the presence of pesticide residues" in Line 764 has been un-bolded for consistency with the rest of the manuscript.

  1. Typographical Errors:

 Typographical errors such as "One grame" have been corrected to "1 g."

  1. Italicization of Latin Names:

"Vitis vinifera L." has been italicized throughout the manuscript for proper scientific notation.

We believe these revisions have significantly improved the manuscript. Thank you again for your invaluable feedback. We are pleased to submit the revised manuscript for your consideration.

Sincerely, 

Ireneusz Ochmian

Reviewer 2 Report

The study compared the chemical, physical, microbiological and functional properties of grape juice from organic and conventional vitis vinifera L. Emphasis was placed on the effect of cultivation methods on the content of health-promoting compounds (such as polyphenols) and potentially harmful substances (such as mycotoxins and pesticide residues) in grape juice. The study provides a scientific basis for understanding the impact of different agricultural practices (organic versus conventional) on the quality of fruits and their processed products, such as grape juice and wine. There are some implications for assessing the health risks and benefits of consuming these products, but the graphic layout and logic of the text are confusing. Here are some suggestions:

Major comments:

1.  The figures need to be improved and provide the reproducible script in GitHub. Such as ImageGP (https://doi.org/10.1002/imt2.5) can generate high quality figures and with reproducible scripts.

2.  The structure of the results appears not in concise. The authors should have presented their findings in around 3 mainly sections will be better.

3.  For statistical analysis, EasyAmplicon have better reproducible steps and scripts is recommended. A network compare with each group is also recommended, such as ggClusterNet, iNAP are good tools for network analysis and visualization.

4.  The pictures of grapes and cultivation gardens in the material method provide a wealth of information, but some of them affect the overall fluency of the article reading. It is recommended to retain and move to the attached drawing.

5.  What does "must" mean in Table 2 and Table 3? The high-frequency word "must" appear in the text, why add this word?

Minor comments:

1.       Line 446, undetected endophytic toxins can be integrated into Table 6 or described in the main text. No need for another subheading.

2.       Line 460, the detection of aflatoxins and ochratoxin gene in Fig1 can be combined into one figure. It's represented by different bars and colors. In addition, words corresponding to genes name need to be italicized.

3.       Why are there two Fig2s in Line 565 and 609? If you add Fig1 and 2 in the materials method, it is even more confusing. Line 609 figures fonts are all crowded together.

4.       Line 733 discusses the effect of viticulture system on the secretory activity of toxigenic fungi. Why is another section of viticulture method on pesticide residues added to Line 764? Consider reorganizing the topic in this paragraph, or separating the two topics.

5.       Line 786, "Fot.3"?? What does Fot mean?

6.       Line 130, is the latin version of the grape variety solaris "Vitis vinifera L. cultivar" in the main text or "vitis vinifera L. farming" in the title? Are they two of the same breeds.

7.       Line 222-227, MeOH, CH3COOH, etc., in the material method. Please unify the text into words, abbreviations or chemical formulas, do not mix. And pay attention to the upper and lower scripts of Arabic numerals.

Author Response

(The authors gave the same response as above.)

Round 2

Reviewer 1 Report

Nothing

The author's manuscript has been improved from last time and my comments are as follows:

1) Figures are presented to illustrate specific experimental results, however, Fig.1; 2 and 7 in the text, what are the authors trying to use to illustrate?

2) Line24, Does 'juice' need to be italicised?

3) Table 2 and 3, A three-line table is required;

4) Fig. 4, A and B should be clearly labelled in the Figure; ‘genesg-1’ should be ‘genes/g’ or ‘genes g-1.

5) Line 867, Reference?

Reviewer 2 Report

The article only describes the correlation without exploring the cause and effect, and does not reach the average level in the high-standard journal.

Data availability is missing. All the data should be open or generate share link for reviews and editor, which is common knowledge in academia. Submitting a manuscript without providing data will not ensure any data basis for this study. So, I recommended that the manuscript be rejected.

Code availability part GitHub link is missing. Do you want to block all the reviews and editor to reanalysis this study? The author did not provide the code to support the analysis of this article, which also violated the basic requirements for submission to this journal. Moreover, the author lists the code paragraphs but deliberately does not provide them, making it impossible to evaluate the scientific attribution.

The article only describes the correlation without exploring the cause and effect, and does not reach the average level in the high-standard journal.

Data availability is missing. All the data should be open or generate share link for reviews and editor, which is common knowledge in academia. Submitting a manuscript without providing data will not ensure any data basis for this study. So, I recommended that the manuscript be rejected.

Code availability part GitHub link is missing. Do you want to block all the reviews and editor to reanalysis this study? The author did not provide the code to support the analysis of this article, which also violated the basic requirements for submission to this journal. Moreover, the author lists the code paragraphs but deliberately does not provide them, making it impossible to evaluate the scientific attribution.